## Replications - stage 2

psychology/cognition

sensory memory, iconic memory, emotion, attention, early sensory processing

**Author for correspondence:**
Florian Kattner
e-mail: kattner@psychologie.tu-darmstadt.de

# Revisiting the prioritization of emotional information in iconic memory

## Florian Kattner[1,2,†] and Alexandra Clausen[1,†]

[1]Institute of Psychology, Technical University of Darmstadt, Alexanderstrasse 10, 64283 Darmstadt, Germany
[2]Institute of Psychology, University of Hamburg, Von-Melle-Park 11, 20146 Hamburg, Germany

  FK, 0000-0003-2124-2829; AC, 0000-0002-3890-3585

In this replication study, the previously reported prioritization of emotional stimuli in iconic memory (Kuhbandner *et al.* 2011. *Psychol. Sci.* **22**, 695–700. (doi:10.1177/0956797611406445)) was reinvestigated. Therefore, recall from iconic memory was measured for sets of visual images that were briefly presented in the periphery of the visual field. Using a partial-report technique, a central arrow presented at varying delays after the images was pointing to the location of the to-be-recalled target. In the direct replication (experiment 1, $n = 41$), participants were asked to verbally report the cued image (note that the entire planned sample size could not be reached owing to the COVID-19 pandemic), and in an extension experiment (experiment 2, $n = 55$), iconic memory was tested using a visual recognition test. Both experiments demonstrated prioritized selection of emotional targets from iconic memory, with higher verbal recall and visual recognition accuracy for negative and positive targets compared to neutral targets. In addition, we found that the presence of emotional distractors in the set interfered with the selection of neutral targets, thus confirming a trend that was observed in the original study. Exponential decay curves further revealed that both target and distractor valence primarily affected initial availability (in case of verbal recall) and attentional selection, whereas the decay of iconic memory with increasing cue delay was less sensitive to the emotional meaning.

## 1. Introduction

Most situations of daily life contain large quantities of visual, auditory, mechanical, and/or chemical information, which simultaneously reach the human sensory receptors. Owing to the limited capacity of both sensory and cognitive systems, selection mechanisms are required to isolate a subset of the information in the environment for further processing. The amount of selective attention that is directed to a stimulus depends on both bottom-up

[†]These authors contributed equally to this study.

and top-down processing such as the salience or relevance of the stimulus, the observer's current goals, and past experiences or learned expectations [1–4]. Moreover, there is plenty of evidence suggesting that selective attention and the degree of sensory processing is influenced also by the emotional significance of a stimulus [5]. For instance, it has been found that threatening or fear-relevant targets (e.g. snakes and spiders) are detected faster than fear-irrelevant targets (e.g. flowers and mushrooms) in a visual search task [6]. Likewise, in an attentional blink paradigm, healthy participants were less likely to miss an aversive stimulus as the second target in a rapid series of visual stimuli, whereas there was no facilitation for aversive targets in patients with bilateral amygdala damage [7]. Moreover, it was found that recognition accuracy for the identity of faces in visual short-term memory is enhanced for angry faces, as compared to neutral and happy faces [8]. Interestingly, the emotional significance of a stimulus was also found to affect early visual processing, as suggested by the enhanced contrast sensitivity in an orientation discrimination task when attention was cued by a fearful face, as compared to when it was cued by a neutral face [9]. This suggests that the prioritization of emotional information may occur already at a very early processing stage prior to attentional selection and in the absence of awareness [10].

In line with these findings, differences between emotional and neutral stimuli were also reported for the read-out of information from iconic memory [11] using an adapted version of Sperling's partial-report paradigm [12]. In this paradigm, the initial availability of information in iconic memory can be measured by asking participants to recall only a subset rather than the full set of target stimuli. Therefore, Sperling [12] presented an auditory cue immediately after the offset of a display of several rows of target letters, telling the participants which row of letters was to be reported. Participants correctly reported almost 100% of the letters in partial report, whereas only about four items could be reported when all rows were to be recalled (whole report). As the to-be-recalled row was not known before presentation of the cue, this finding suggests that all items must be initially available in iconic memory. However, owing to the limited capacity of visual short-term memory [13], only a small number of items (e.g. four letters) can be selected from iconic memory for recall and further processing. Moreover, the information available in iconic memory was found to decay rapidly with increasing delay of the cue, suggesting that the time window for read-out is very short [14]. Kuhbandner *et al.* [11] found that, while both emotional and neutral information were initially equally likely to be available in iconic memory, both the selection of information from iconic memory (enabling transfer to visual short-term memory and verbal recall) and the decay of information that was initially available in iconic memory was influenced by the emotional meaning of the stimuli. Specifically, eight images were presented simultaneously in a circular array for a brief period of time (129 ms) and participants were asked to verbally report a single target item, as indicated by an arrow cue that was presented after a variable delay. Immediate recall accuracy was higher when the target item was emotional (threatening or positive) than when it was a neutral image (recall accuracy of approx. 83% for emotional targets versus 65% for neutral targets with the shortest cue delays), indicating that emotionally significant stimuli are more likely to be selected for read-out from iconic memory. In addition, the authors reported slower decay of information with increasing cue delay for threatening targets, as compared to neutral and positive targets, suggesting that the time window for read-out also depends on the emotional significance. Hence, the results suggest that both negative and positive information is prioritized in terms of the attentional selection from iconic memory, whereas only negative information is also prioritized in terms of a slower decay of information (i.e. prolonged availability in iconic memory). The authors argued that the valence asymmetry with regard to the temporal dynamics of iconic memory might be owing to differences in arousal between the negative and positive stimuli used in that study (e.g. a spider versus a heart).

While the access to iconic memory differed between emotional and neutral targets, the authors found no significant effects of the emotional significance of non-target stimuli [11]. That is, although there was a trend, the presence of emotional non-targets (as compared to neutral non-targets) did not reliably interfere with recall accuracy of neutral targets. This might indicate that emotional information may only facilitate target processing, but not lead to distraction at the level of iconic memory. By contrast, in many other studies, irrelevant emotional information was found to disrupt processing of neutral target information at post-attentional stages [15–18].

The present study is an attempt to replicate and extend the theoretically important findings reported by Kuhbandner *et al.* [11]. Considering that the study demonstrated large effects of emotional significance at very early stages of visual processing, it is surprising that it received relatively little attention in the literature (only 19 citations according to Web of Science, 28 August 2019).[1] It could be the case that

---

[1]During the process of preparing the manuscript, we have become aware of a previous unpublished attempt to replicate the findings of that study (K. Wong, D. Cervone: *Iconic memory for affective stimuli: a replication*).

this neglect is owing to the diverging effects of emotional significance on (i) the immediate selection of information from iconic memory, (ii) the temporal properties of iconic memory, and (iii) the disruption of neutral stimulus processing. Specifically, both negative and positive information was prioritized for immediate read-out, whereas only negative information was prioritized in terms of a slower decay, and neither positive nor negative distractors interfered with the read-out of neutral information from iconic memory. Hence, it could be argued that only 1.5 of 3 hypotheses were supported by the data. In order to substantiate the empirical basis for emotional effects at the level of iconic memory, it seems crucial to replicate the findings in experiment 1 using the same paradigm and stimuli as Kuhbandner *et al.* [11], but with increased statistical power and a larger number of repetitions per experimental condition (i.e. increasing within-subjects power).

In addition to the direct replication with verbal recall from iconic memory, an extension experiment was conducted with the verbal recall task being replaced by a visual recognition test in which participants are not required to retrieve a verbal label of the stimulus from long-term memory. More specifically, participants of experiment 2 were asked to choose the visual target from a constant selection of images shown at the bottom of the screen. In a verbal recall task, the iconic information needs to be transferred to verbal short-term memory (i.e. by retrieving a verbal label from long-term memory and generating a phonological code enabling the articulatory response) which may be accompanied by a loss of visual detail information. Moreover, verbal recall accuracy may be affected by individual differences in verbal fluency or phonological capacity.

By contrast, visual recognition avoids the use of a phonological short-term memory system during the response stage, and it may be better suited to test the read-out of visual information held in iconic memory. However, it could be argued that the visual response options shown in a recognition task may interfere with the information held in visual working memory and cause intrusions. We do not consider this to be very likely if only a single target item needs to be held in visual short-term memory at the time of the recognition test (i.e. when the response options are shown). Moreover, it is unlikely that these intrusions occur at the level of iconic memory as the selection of information from iconic memory is expected to occur very rapidly and before participants can make an eye movement to the bottom of the screen. To be more specific, participants are supposed to read the information from the periphery of the iconic memory image while fixating the centre of the screen where the cue is presented. The target information from iconic memory then needs to be transferred immediately to visual short-term memory before visual attention is moved to the response options, which are shown at the bottom of the screen (it has been found with visual search paradigms that the matching of a single template in a visual recognition task is a very fast and efficient process; e.g. [19,20]). Therefore, the selection of information from iconic memory is likely to be completed at the time when attention is directed to the response options, and any possible intrusions must occur at the level of visual short-term memory for which the load is very low. Therefore, interference in visual short-term memory with a single to-be-remembered target item should be (i) negligible owing to the very low memory load (one item), and (ii) not relevant for measuring the selection of information from iconic memory (which should be completed at this stage). We thus expect to replicate the prioritization of emotional information in experiment 2 using a visual recognition task. Replacing the verbal responses with a visual recognition test provides a valuable extension with the opportunity to demonstrate the generalizability of the emotional effects on iconic memory reported by Kuhbandner *et al.* [11].

In both experiments, each participant completed twice the number of trials compared to the original study (i.e. 400) in order to assess possible attentional habituation effects [21]. Therefore, in addition to the direct replication in the first 200 trials, the emotional effects on iconic memory were also contrasted between the first and second block of 200 trials. In the original study [11], exponential decay curves could not be fitted to the data of individual participants owing to the low number of repetitions per conditions (and distortions of parameter estimates). In the present study, decay curves were fitted for each individual by collapsing data in both 200-trial blocks in order to test for possible effects of cue delay and emotional meaning on the parameter estimates of the decay curves.

## 2. Pre-registration

This article received in-principle acceptance (IPA) as a results-blind replication at Royal Society Open Science on 10 September 2019. Following IPA, the accepted Stage-1 version of the study protocol, not including results and discussion, was pre-registered on the Open Science Framework (OSF):

# 3. Experiment 1: direct replication

## 3.1. Methods

### 3.1.1. Participants

A power analysis for the main effect of emotional significance of the immediate read-out of information from iconic memory at the shortest cue delay revealed a sample size of 68 participants to be required for an effect size of $d_z = 0.36$ ($M = 0.83$; s.e.m. $= 0.075$ for emotional items versus $M = 0.65$; s.e.m. $= 0.075$, based on fig. 1b in [11]) to be detected with a statistical power of 90% (i.e. $1-\beta = 0.90$; $\alpha = 0.05$). Therefore, we planned to recruit a total of 68 participants at the campus of Technical University of Darmstadt, but owing to the COVID-19 pandemic, it was not possible to continue data collection with human participants in the laboratory after 13 March 2020. The final sample thus consisted of $n = 41$ individuals who had participated prior to the outbreak (28 women, 13 men). Ages ranged between 18 and 56 years ($M = 23.7$; s.d. $= 7.4$). All participants agreed to participate without financial compensation and eligible participants were compensated with course credit. A sensitivity analysis (using the {pwr} package for R, [22]) revealed that an effect size of $d_z = 0.46$ or larger can thus be detected in a paired $t$-test (one-sided) with a statistical power of 90% ($\alpha = 0.05$), whereas the previously reported effect size of $d_z = 0.36$ could still be detected with a statistical power of 74%.

### 3.1.2. Apparatus and stimuli

The experiment was conducted in a dimly lit, single-walled sound attenuated listening booth. Visual stimuli were presented on a 17-inch CRT monitor (Dell E773p, 60 Hz), and participants were seated at approximately 40 cm viewing distance. The experimental routines were programmed in MATLAB (Mathworks, Natick, MA) using the Psychophysics toolbox 3.0 extensions [23–25].

The identical set of 12 monochrome drawings of animals and objects as in the original study [11] was used to test iconic memory. Four drawings were categorized to be neutral (tree, fish, dog and deer), four were positive (heart, butterfly, kiss and jubilant person) and four were negative or threatening images (spider, scorpion, gun and skull). To confirm the differential emotional meaning of the drawings, an independent sample of 20 participants was asked to rate both the valence and the arousal of the twelve drawings on 7-point Likert scales (1 = negative/low arousal; 7 = positive/high arousal). The valence ratings differed significantly between neutral ($M = 4.71$; s.d. $= 0.65$), positive ($M = 5.48$; s.d. $= 0.70$), and negative drawings ($M = 2.22$; s.d. $= 0.68$), $F_{2,38} = 128.34$; $p < 0.001$; $\eta_G^2 = 0.81$. Pairwise $t$-tests corrected for multiple comparisons [26] revealed that all three categories differed significantly from each other (all $p$s $< 0.001$). Moreover, the three categories of drawings also differed in terms of the arousal ratings, $F_{2,38} = 5.09$; $p = 0.01$; $\eta_G^2 = 0.10$, with higher arousal ratings for negative drawings ($M = 4.54$; s.d. $= 1.39$) than for positive ($M = 3.95$; s.d. $= 1.26$) and neutral drawings ($M = 3.56$; s.d. $= 1.01$). While there were significant differences between negative and neutral ($p < 0.001$) and between negative and positive drawings ($p = 0.04$), positive and neutral drawings did not differ in terms of arousal ratings ($p = 0.15$). We further note that there was a small but non-significant physical difference in 8-bit luminance values of the drawings in the neutral ($M = 195.6$; s.d. $= 15.9$), positive ($M = 200.5$; s.d. $= 19.3$), and negative category ($M = 211.4$; s.d. $= 12.2$), $F_{2,9} = 1.01$; $p = 0.40$; $\eta_G^2 = 0.18$.

### 3.1.3. Procedures

All procedures were conducted strictly in accordance with the Declaration of Helsinki, and the study was approved by the ethics committee of Technical University of Darmstadt on 13 February 2017 (EK 04/2017). All participants gave written informed consent prior to participating.

The experiment consisted of a practice phase and a main phase. The practice phase consisted of 60 trials in which stimuli from a different set of 12 neutral visual stimuli showing transportation objects (e.g. car, bicycle, boat, and plane) were presented. Besides the stimuli and the presentation of feedback in the practice phase, the procedure was identical for the practice and the main phase. For the main phase, a 5 (delay: 17, 68, 221, 493 or 1003 ms) × 5 (emotion of target and distractors: neutral-neutral, positive-neutral, negative-neutral, neutral-positive and neutral-negative) experimental design was

implemented with 16 repetitions of each condition, resulting in a total of 400 trials. The order of trials of the main phase was randomized within each full block of 25 trials (containing each condition once). A short break could be taken after every fourth block (i.e. 100 trials). At the beginning of both the practice phase and the main phase, participants were asked to watch the 12 images from the respective phase on a sheet of paper for 60 s.

Each trial started with a black fixation cross in the centre of the white screen for 750 ms before eight stimuli were presented simultaneously for 136 ms (eight frames), evenly spaced on an invisible circle around the fixation point at an eccentricity of 6°. Four different stimuli were drawn on each trial with each stimulus being shown twice at random locations. On the neutral-distractor trials, three neutral stimuli were presented as distractors, and an additional either neutral (neutral-neutral), positive (positive-neutral) or negative (negative-neutral) stimulus was shown as the target. On the emotional-distractor trials, a single neutral stimulus was shown as the target and either three different positive (neutral-positive) or three different negative stimuli (neutral-negative) were presented as the distractors. After the variable delay showing a white blank screen, a black arrow was presented pointing in the direction of the to-be-recalled target stimulus. Participants were now asked to verbally report the image that was presented at the cued location. The response was entered manually into a computer by the experimenter [11,27,28]. During the practice phase, the correct image was presented as visual feedback on the screen for 500 ms. No feedback was given during the main phase of the experiment.

### 3.1.4. Data analysis

The accuracy recall responses were first tested against chance level (8.3% correct) for each participant using one-sample $t$-tests, and data were not included for the analyses if the participant's accuracy did not differ significantly from chance ($p > 0.05$).

To test for emotional effects on iconic memory, 3 (target or distractor valence) × 5 (cue delay) repeated-measures ANOVAs were conducted on the participants' average recall accuracy at the five different cue delays for the different target and distractor valence conditions. As the planned sample size of $n = 68$ could not be reached, additional bootstrapping analyses were conducted to estimate the 90% confidence intervals (CI) of the effect size estimates with the targeted sample size (i.e. using 100 independent resamples each consisting of 68 subjects randomly drawn with replacement from the data of the first and second block). Individual contrasts were tested using pairwise $t$-tests corrected for multiple comparisons by controlling for the false discovery rate using the method suggested by Benjamini & Hochberg [26]. The initial analysis was conducted with the first 200 trials as in the original study in order to avoid habituation effects. In addition, to test for possible attentional habituation effects, we contrasted the effects of valence and cue delay between the first and second block of 200 trials.

To describe the temporal dynamics of iconic memory, exponential decay functions, $p(t) = \alpha e^{-t/\tau} + \beta$ [27,28], were fitted to the average verbal recall accuracy using an adaptive nonlinear least-squares algorithm [29]. In this function, $p(t)$ refers to the probability of a correct response at cue delay $t$. The parameter $\alpha$ represents the initial availability of visual stimulus information in iconic memory (which does not necessarily get transferred to short-term memory), the parameter $\beta$ represents the amount of information that is selected and transferred to visual short-term memory, and the parameter $\tau$ represents the temporal constant (or duration) of iconic memory characterizing the degree of exponential decay of initially available information. In line with the original study [11], the attentional selection parameter $\beta$ is expected to be enhanced for emotional targets, whereas the decay parameter $\tau$ may be enhanced only for negative (threatening) targets. In addition to the original study, decay functions were also fitted to individual's average recall accuracy across all 400 trials using a similar algorithm [30,31] in order to test for statistical differences in parameters between cue delay and valence conditions.

In addition, Bayes factors (BF) (using the {BayesFactor} package for R [32]) were calculated to obtain evidence for (i) a model 1 with two independent fixed effects for target/distractor valence and delay and (ii) a model 2 assuming an additional valence × delay to account for recall accuracy, relative to a model with only a fixed effect of delay (and a random effect of valence).

## 3.2. Results

### 3.2.1. Effects of target valence

Participants accuracy to verbally recall positive, negative, or neutral targets in the first and second 200-trial block of the iconic memory task is illustrated in figure 1*a,b*, respectively. Consistent with the original

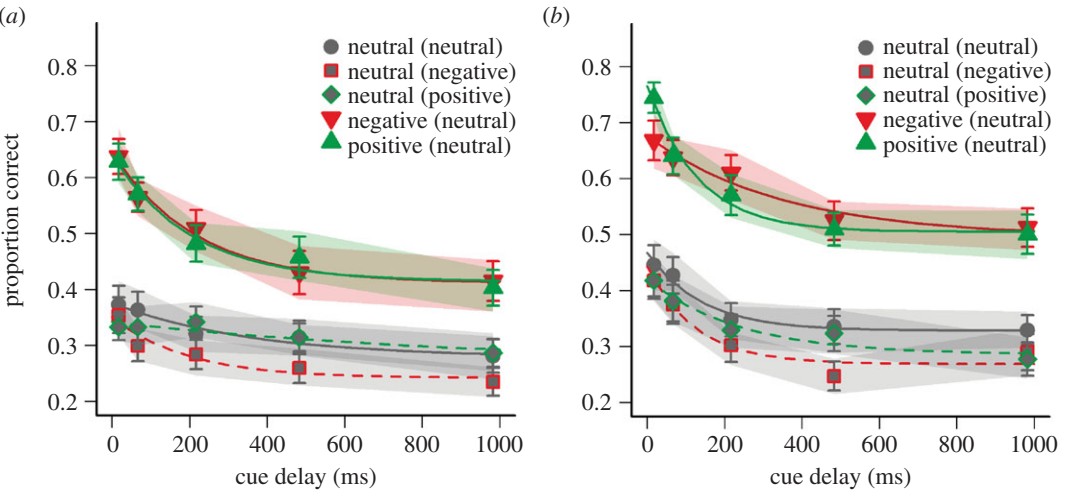

**Figure 1.** Mean verbal recall accuracy for neutral, positive and negative targets as a function of the cue delay in the first (a) and second (b) 200-trial block of experiment 1. Positive and negative targets were to be selected against neutral distractor, and neutral targets were to be selected against neutral, positive, or negative distractors (distractor valence indicated in the parentheses of the legend). Error bars depict standard errors of the mean. Solid and dashed lines represent exponential decay functions fitted to recall accuracy. The shaded areas indicate 90% CI of mean accuracy for a bootstrapped sample size of 68 participants (100 resamples).

study, it can be seen that accuracy in the first block of 200 trials was higher on average for positive ($M = 0.51$; s.d. $= 0.14$) and negative targets ($M = 0.51$; s.d. $= 0.13$) than for neutral targets ($M = 0.33$; s.d. $= 0.13$) which were to be selected against neutral distractors. A 3 (target valence: positive, negative, neutral; all with neutral distractors) $\times$ 5 (delay: 17, 68, 221, 493, 1003 ms) repeated-measures ANOVA on accuracy in the first block confirmed this main effect of target valence, $F_{2,80} = 43.18$; $p < 0.001$; $\eta_G^2 = 0.15$ (bootstrapped 90% CI $= (0.11; 0.19)$). Pairwise $t$-tests corrected for multiple comparisons [26] revealed that accuracy for positive and negative targets differed significantly from neutral targets (with neutral distractors), $p < 0.001$, whereas there was no difference between positive and negative targets, $p = 0.86$. The ANOVA further revealed a significant main effect of delay in the first block, $F_{4,160} = 19.10$; $p < 0.001$; $\eta_G^2 = 0.09$ (CI $= (0.07; 0.12)$), suggesting a gradual decay of information available in iconic memory from the shortest ($M = 0.55$; s.d. $= 0.15$) to the longest delay ($M = 0.37$; s.d. $= 0.14$). However, the ANOVA did not reveal a significant interaction between target valence and delay, $F_{8,320} = 1.60$; $p = 0.12$; $\eta_G^2 = 0.01$ (CI $= (0.01; 0.03)$), indicating that the slope of decay curves did not differ between the three target valence conditions (compare solid lines in figure 1a). An additional BF analysis [32] revealed that both models 1 and 2 including target valence as a fixed effect were much more likely than a null model with only delay as a fixed effect ($BF_{10} \approx 1.3 \cdot 10^{23}$; $BF_{20} \approx 6.6 \cdot 10^{21}$). More importantly, the data of the first block provide more evidence for a model 1 with two independent fixed effects for target valence and delay than for a model assuming a target valence $\times$ delay interaction ($BF_{12} \approx 20.0$), suggesting that emotional targets were more likely to be selected from iconic memory than neutral targets, whereas there is less evidence for differential decay of neutral and emotional information in iconic memory.

The same analysis of target valence effects was also conducted for the second block of 200 trials (figure 1b), revealing again a main effect of target valence, $F_{2,80} = 66.80$; $p < 0.001$; $\eta_G^2 = 0.20$ (CI $= (0.16; 0.25)$), as well as a main effect of delay, $F_{4,160} = 26.70$; $p < 0.001$; $\eta_G^2 = 0.09$ (CI $= (0.07; 0.12)$), but no interaction $F_{8,320} = 1.28$; $p = 0.26$; $\eta_G^2 = 0.01$ (CI $= (0.01; 0.03)$). Again, the corrected pairwise $t$-tests revealed that accuracy for positive and negative targets differed significantly from neutral targets in the second block, $p < 0.001$, but there was no difference between positive and negative targets, $p = 0.89$. Consistent with these results, a BF analysis based on the data in the second block revealed again that model 1 with two independent fixed effects of target valence and delay was more likely than a model with the interaction term as a fixed effect ($BF_{12} \approx 30.2$). Again, both models were much more likely than a null model without target valence as a fixed effect ($BF_{10} \approx 1.4 \cdot 10^{35}$; $BF_{20} \approx 4.5 \cdot 10^{33}$).

Contrasting the two blocks of the experiment revealed a general practice effect with accuracy being significantly higher in the second block ($M = 0.52$; s.d. $= 0.12$) than in the first block ($M = 0.45$; s.d. $= 0.11$), $F_{1,40} = 44.44$; $p < 0.001$; $\eta_G^2 = 0.03$, but there were no significant interaction with block, $F < 1.18$; $p > 0.31$, suggesting that the effects of target valence remained stable across the entire 400 trials of the

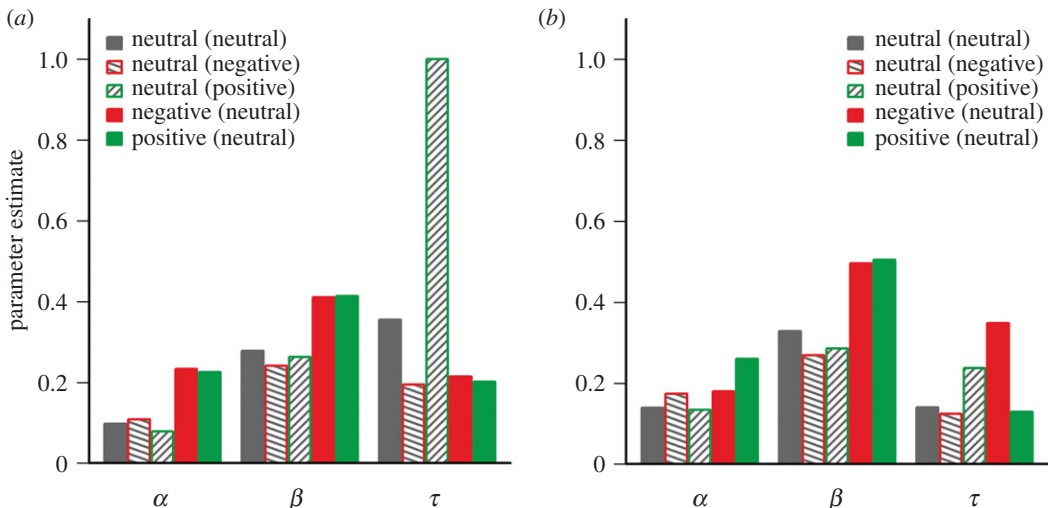

**Figure 2.** Parameter estimates of exponential decay functions fitted to the aggregated data in the first (*a*) and second (*b*) block for the initial availability of stimulus information in iconic memory ($\alpha$), the amount of information that was transferred from iconic memory to visual short-term memory ($\beta$), and the degree of exponential decay of the initially available information ($\tau$; with higher values indicating less decay).

task. Furthermore, it is interesting to note that the interaction between target valence and delay was significant when the data were collapsed across the two blocks, $F_{8,320} = 2.13$; $p = 0.03$; $\eta_G^2 = 0.01$.

Exponential decay functions were fitted to the recall accuracy in order to contrast the decay functions between different valence conditions. The best fitting functions in the first and second block are shown in figure 1*a*,*b*, and the respective parameter estimates are visualized in figure 2*a*,*b*. Consistent with the original study [11], the initial availability parameter $\alpha$ appears to be enhanced for emotional target in particular for the first block of 200 trials. In addition, it can be seen that the selection parameter $\beta$ was higher for positive and negative targets than for neutral targets in both blocks. Interestingly, the temporal decay parameter $\tau$ was higher for negative targets than for positive targets only in the second block of 200 trials, indicating slower decay of negative information. By contrast, in the first block, higher values of $\tau$ were observed for neutral targets (in particular those paired with positive distractors), suggesting that the valence of distractors may have had a stronger effect on the decay curves in the earlier trials.

To test for possible effects of target valence on parameter estimates, exponential decay functions were also fitted to the 400 trials of each individual participant. For the initial availability parameter $\alpha$, there was a significant difference between positive ($M = 0.36 \pm 0.03$), negative ($M = 0.33 \pm 0.04$) and neutral targets with neutral distractors ($M = 0.21 \pm 0.03$), $F_{2,80} = 4.35$; $p = 0.02$; $\eta_G^2 = 0.08$. Moreover, there was also a significant effect of target valence on the selection parameter $\beta$, $F_{2,80} = 7.80$; $p < 0.001$; $\eta_G^2 = 0.09$, with prioritized selection of negative ($M = 0.40 \pm 0.03$) and positive targets ($M = 0.38 \pm 0.03$) compared to neutral targets ($M = 0.27 \pm 0.02$). However, there was no effect of target valence on the decay parameter $\tau$, $F_{2,80} = 0.28$; $p = 0.75$; $\eta_G^2 < 0.01$, suggesting that emotional targets were not characterized by a slower decay.

It is further important to note that the verbal recall advantage of negative targets (collapsed across both blocks) differed significantly between the four exemplars of negative images (spider: $M = 0.65$; s.d. = 0.16, scorpion: $M = 0.58$; s.d. = 0.15, pistol: $M = 0.56$; s.d. = 0.19, skull: $M = 0.41$; s.d. = 0.18), $F_{3,120} = 22.03$; $p < 0.001$; $\eta_G^2 = 0.21$. Likewise, recall accuracy also differed significantly between the positive (heart: $M = 0.68$; s.d. = 0.16, kiss: $M = 0.57$; s.d. = 0.16, butterfly: $M = 0.48$; s.d. = 0.18, jubilant: $M = 0.48$; s.d. = 0.23), $F_{3,120} = 17.47$; $p < 0.001$; $\eta_G^2 = 0.17$, and neutral images (tree: $M = 0.48$; s.d. = 0.17, dog: $M = 0.42$; s.d. = 0.17, fish: $M = 0.36$; s.d. = 0.18, deer: $M = 0.15$; s.d. = 0.12), $F_{3,120} = 48.73$; $p < 0.001$; $\eta_G^2 = 0.37$.

### 3.2.2. Effects of distractor valence

It can also be seen in figure 1*a*,*b* that accuracy to recall neutral targets was lower when they were to be selected against emotional distractors than when neutral distractors were presented, in particular for negative distractors and in the second block. A 3 (distractor valence: positive, negative, neutral) × 5

(delay) repeated-measures ANOVA on identification accuracy for neutral targets in the first block of 200 trials confirmed this main effect of distractor valence, $F_{2,80} = 4.08$; $p = 0.02$; $\eta_G^2 = 0.01$ (CI = (0.005; 0.02)), suggesting that emotional distractors disrupted the selection of neutral information from iconic memory (positive distractors: $M = 0.32$; s.d. = 0.11, negative distractors: $M = 0.29$; s.d. = 0.10), as compared to neutral distractors ($M = 0.33$; s.d. = 0.13). Alpha-corrected pairwise comparisons revealed a significant contrast between negative and neutral distractors, $p = 0.02$, as well as between positive and negative distractors, $p = 0.02$, but not between positive and neutral distractors, $p = 0.61$. The analysis also revealed a main effect of delay, $F_{4,160} = 5.79$; $p < 0.001$; $\eta_G^2 = 0.03$ (CI = (0.025; 0.04)), but no interaction, $F_{8,320} = 0.55$; $p = 0.82$; $\eta_G^2 < 0.01$ (CI = (0.004; 0.02)). The results were confirmed by a BF analysis, showing that a model 1 with two fixed effects of distractor valence and delay was more likely than a model 2 which also included an interaction term (BF$_{12} \approx$ 188.8). However, based on the data in the first block, model 1 was about as likely as a null model with only delay as a fixed effect (BF$_{10} \approx 1.3$), whereas model 2 was less likely than the null model (BF$_{10} \approx 0.007$).

The same effects were confirmed in the second block, with higher accuracy for neutral targets that were selected against other neutral items ($M = 0.38$; s.d. = 0.13) than for neutral targets that were presented together with positive ($M = 0.35$; s.d. = 0.14) or negative distractors ($M = 0.33$; s.d. = 0.12), $F_{2,80} = 4.43$; $p = 0.02$; $\eta_G^2 = 0.01$ (CI = (0.004; 0.02)). Again there was a significant contrast between negative and neutral distractors, $p = 0.006$. However, there was no difference between positive and negative distractors, $p = 0.23$, and a marginally significant difference between positive and neutral distractors, $p = 0.09$. There was also a main effect of delay, $F_{4,160} = 14.56$; $p < 0.001$; $\eta_G^2 = 0.06$ (CI = (0.05; 0.10)), but no interaction, $F_{8,320} = 0.49$; $p = 0.87$; $\eta_G^2 < 0.01$ (CI = (0.004; 0.02)), suggesting that the valence of non-target information in iconic memory did not affect the decay of neutral information in iconic memory with increasing cue delay. Again, this pattern of results was confirmed by a BF analysis, indicating that a model 1 with fixed effects for distractor valence and delay was more likely than the interaction model 2 (BF$_{12} \approx 237.3$). Model 1 was only slightly more likely than the null model without distractor valence as a fixed effect (BF$_{10} \approx 1.6$), whereas model 2 was less likely than the null model (BF$_{20} \approx 0.007$).

In general, accuracy of neutral targets was higher in the second block ($M = 0.35$; s.d. = 0.11) than in the first block ($M = 0.31$; s.d. = 0.10), $F_{1,40} = 13.96$; $p < 0.001$; $\eta_G^2 = 0.01$. In addition, there was a significant interaction between block and delay, $F_{4,160} = 2.66$; $p = 0.03$; $\eta_G^2 = 0.01$, indicating that the decay curves were steeper in the second block (from $M = 0.43$; s.d. = 0.15 at the shortest delay to $M = 0.30$; s.d. = 0.12 at the longest delay) than in the first block (from $M = 0.35$; s.d. = 0.15 at the shortest delay to $M = 0.27$; s.d. = 0.10 at the longest delay). There were no other interactions with block, $F < 1$; $p > 0.64$.

The individual parameter estimates of exponential decay curves fitted to the recall accuracy for neutral targets (across both blocks) were also tested for possible effects of distractor valence. For the initial availability parameter $\alpha$ of neutral targets, there was no significant difference between neutral, positive, and negative distractors, $F_{2,80} = 0.95$; $p = 0.30$ $\eta_G^2 = 0.01$. Likewise, there was no differences in the selection parameter $\beta$, $F_{2,80} = 1.79$; $p = 0.17$; $\eta_G^2 = 0.02$, and the temporal decay parameter $\tau$, $F_{2,80} = 0.28$; $p = 0.75$; $\eta_G^2 < 0.01$, indicating that distractor valence did not reliably affect the shape of the exponential decay curves of neutral targets.

## 3.3. Discussion

Experiment 1 demonstrated that the effects of emotional valence on the selection of information from iconic memory [11] can be replicated. Specifically, we found that both positive and negative targets (e.g. schematic shapes of a spider, a pistol, or a heart) were recalled at higher accuracy than emotionally neutral targets (e.g. the shape of a tree or a fish). In addition, we found that verbal recall accuracy of neutral targets was further impaired by the presence of negative emotional distractors, thus confirming the trend of distractor interference which was reported in the original study [11]. Moreover, exponential decay functions fitted to the individuals' verbal recall accuracy reveal that the valence of to-be-recalled items in iconic memory affected their initial availability as well as the subsequent selection of the item for further processing, but not the decay of the trace in iconic memory with increasing delay of the recall cue. In a second experiment, we aim to test the generalizability of these effects by measuring the effects of emotional targets and distractors on the selection from iconic memory using a non-verbal, visual recognition task instead of verbal recall.

# 4. Experiment 2: extension

## 4.1. Methods

### 4.1.1. Participants

A slightly lower statistical power of $1 - \beta = 0.8$ was used for the extension experiment, requiring a minimum of 50 participants for the effect size observed by Kuhbandner *et al.* [11]. An additional sample of 55 individuals (35 women) was thus recruited at the campus of the Technical University of Darmstadt to participate in experiment 2 (we note that the data collection for experiment 2 has been completed prior to the COVID-19 pandemic). Ages ranged between 18 and 86 years ($M = 28.8$; s.d. = 13.9). As in experiment 1, participants were compensated with course credit.

### 4.1.2. Apparatus, stimuli and procedure

The same apparatus, stimuli and ethics approval as in experiment 1 were used for experiment 2. The procedure was identical to experiment 1 except for the verbal recall task being replaced by a visual recognition test. The experiment again consisted of a practice phase of 60 trials (showing different stimuli) and a main phase of 400 trials. As in experiment 1, eight images were presented in the periphery and an arrow was presented after a variable delay. However, together with the presentation of the arrow cue, all 12 drawings were presented at the bottom of the screen (in the same order on every trial for a given participant), and participants were asked to click on the drawing that was shown at the position that was indicated by the arrow. Hence, participants were not asked to make a verbal response, but to click on a response icon based on purely visual recognition. During the practice phase, short text feedback (saying 'correct' or 'wrong' in German) was presented immediately after the response for 500 ms, whereas no feedback but a 500 ms inter-trial interval showing a blank screen was presented during the main phase.

### 4.1.3. Data analysis

The visual recognition accuracy was analysed in the same way as verbal recall accuracy in experiment 1 using both frequentist ($3 \times 5$ repeated-measures ANOVA) and Bayesian statistics. In addition, exponential decay curves were fitted to visual recognition accuracy as a function of the cue delay.

## 4.2. Results

The data for two female participants were not included in the analysis because recognition accuracy in the iconic memory task was not significantly different from chance level of 8.33% correct (8.75%; $p = 0.77$ and 9.00%; $p = 0.64$, respectively). For the remaining participants (age range: 18–61 years; $M = 27.6$; s.d. = 11.7), average visual recognition accuracy in the first and second 200-trial block is shown in figure 3*a,b* as a function of the delay of the cue and for the five different conditions of target and distractor valence.

### 4.2.1. Effects of target valence

Visual recognition accuracy in the first block was higher for positive ($M = 0.43$; s.d. = 0.16) and negative targets ($M = 0.43$; s.d. = 0.16) than for neutral targets ($M = 0.29$; s.d. = 0.10; neutral distractors), and a 3 (target valence) $\times$ 5 (delay) repeated-measures ANOVA confirmed this observation with a main effect of target valence, $F_{2,104} = 43.44$; $p < 0.001$; $\eta_G^2 = 0.09$. Pairwise *t*-tests corrected for multiple comparisons [26] revealed a significant contrast between positive and neutral targets, $p < 0.001$, and between negative and neutral targets, $p < 0.001$, but not between positive and negative targets, $p = 0.69$. The analysis further revealed a significant main effect of delay, $F_{4,208} = 17.81$; $p < 0.001$; $\eta_G^2 = 0.06$, indicating the decay of information in iconic memory from the shortest ($M = 0.42$; s.d. = 0.14) to the longest delay ($M = 0.27$; s.d. = 0.09), as well as a significant interaction between target valence and delay, $F_{8,416} = 2.99$; $p = 0.003$; $\eta_G^2 = 0.02$, suggesting that the slope of the decay curves differed between the three target valence conditions (cf. figure 3*a*).

An additional BF analysis based on the data in the first block of experiment 2 revealed that model 2 assuming independent fixed effects of target valence and delay as well as an interaction term was about as likely as (or even slightly more likely than) a model with only the two fixed effects ($BF_{12} \approx 0.9$).

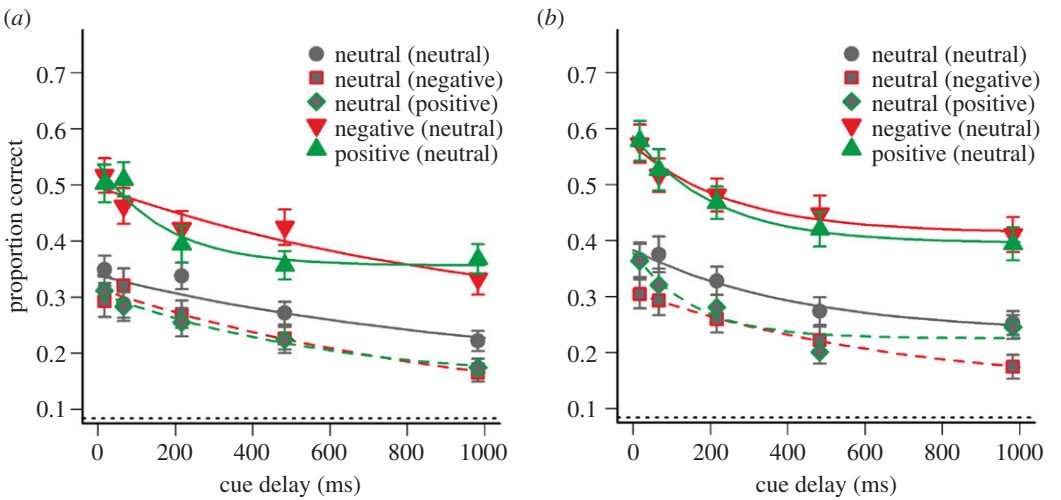

**Figure 3.** Mean visual recognition accuracy of neutral, positive, and negative targets as a function of the cue delay in the first (a) and second (b) block of experiment 2. Positive and negative targets were to be selected against neutral distractor, and neutral targets were to be selected against neutral, positive, or negative distractors (distractor valence indicated in the parentheses of the legend). Error bars depict standard errors of the mean. Solid and dashed lines represent exponential decay functions fitted to recall accuracy. The dotted line at the bottom represents the chance level for target recognition.

However, both models including target valence as a fixed effect were much more likely than a null model with only delay as a fixed effect ($BF_{10} \approx 1.7 \cdot 10^{21}$; $BF_{20} \approx 1.9 \cdot 10^{21}$).

For visual recognition accuracy in the second block of 200 trials (figure 3b), the ANOVA also revealed a significant main effect of target valence, $F_{2,104} = 53.57$; $p < 0.001$; $\eta_G^2 = 0.11$, with better recognition of positive ($M = 0.48$; s.d. $= 0.18$) and negative targets ($M = 0.49$; s.d. $= 0.18$), as compared to neutral targets ($M = 0.32$; s.d. $= 0.13$), $p < 0.001$, but no difference between positive and negative targets, $p = 0.52$. There was also a main effect of delay, $F_{4,208} = 19.62$; $p < 0.001$; $\eta_G^2 = 0.06$. However, in contrast to the first block, there was no interaction between target valence and delay, $F_{8,416} = 0.67$; $p = 0.72$; $\eta_G^2 < 0.01$, indicating that the valence-dependent differences with regard to the slope of the decay curves (i.e. the prolonged availability of negative targets, see figure 3a) was restricted to the first 200 trials. Consistent with this result (and with experiment 1), the BF analysis based on the data in the second block confirmed that model 1 with two independent fixed effects of target valence and delay was more likely than model 2 which also included the interaction term ($BF_{21} \approx 359.2$), and both models were more likely than the null model not including target valence ($BF_{10} \approx 9.6 \cdot 10^{28}$; $BF_{20} \approx 2.7 \cdot 10^{26}$).

Contrasting the two blocks revealed a general improvement of visual recognition accuracy from the first ($M = 0.38$; s.d. $= 0.12$) to the second block ($M = 0.43$; s.d. $= 0.15$), $F_{1,52} = 17.03$; $p < 0.001$; $\eta_G^2 = 0.01$, but no interactions with block, $F < 1.32$; $p > 0.23$, suggesting that the effects of target valence and delay did not differ between blocks.

Again, exponential decay curves were fitted to recognition accuracy. The best fitting functions in the two blocks of experiment 2 are illustrated in figure 3a,b, and the corresponding parameter estimates are shown in figure 4a,b, respectively. As can be seen, in particular the selection parameter $\beta$ was higher for emotional targets than for neutral targets in both blocks, whereas no such difference is evident for the initial availability parameter $\alpha$ and the temporal parameter $\tau$. For the purpose of statistical comparisons, decay functions were also fitted to the data of each participant (collapsed across both blocks). The analysis confirmed a significant effect of target valence on the selection parameter $\beta$, $F_{2,104} = 19.33$; $p < 0.001$; $\eta_G^2 = 0.13$, with higher estimates for positive ($M = 0.35 \pm 0.02$) and negative targets ($M = 0.34 \pm 0.03$) than for neutral targets ($M = 0.21 \pm 0.02$; neutral distractors). By contrast, there was no significant effect of target valence on the initial availability parameter $\alpha$, $F_{2,104} = 2.11$; $p = 0.13$; $\eta_G^2 = 0.02$, and the temporal parameter $\tau$, $F_{2,104} = 0.51$; $p = 0.60$; $\eta_G^2 < 0.01$.

As in experiment 1, we note that the advantage in visual recognition performance for negative targets (collapsed across both blocks) differed significantly between the four negative images (spider: $M = 0.58$; s.d. $= 0.20$, pistol: $M = 0.48$; s.d. $= 0.21$, scorpion: $M = 0.44$; s.d. $= 0.19$, skull: $M = 0.33$; s.d. $= 0.19$), $F_{3,156} = 30.48$; $p < 0.001$; $\eta_G^2 = 0.17$. Likewise, visual recognition accuracy also differed significantly between the various positive (heart: $M = 0.56$; s.d. $= 0.20$, kiss: $M = 0.48$; s.d. $= 0.22$, jubilant: $M = 0.41$; s.d. $= 0.22$, butterfly: $M = 0.35$; s.d. $= 0.16$), $F_{3,156} = 24.88$; $p < 0.001$; $\eta_G^2 = 0.14$, and neutral images (tree: $M = 0.44$;

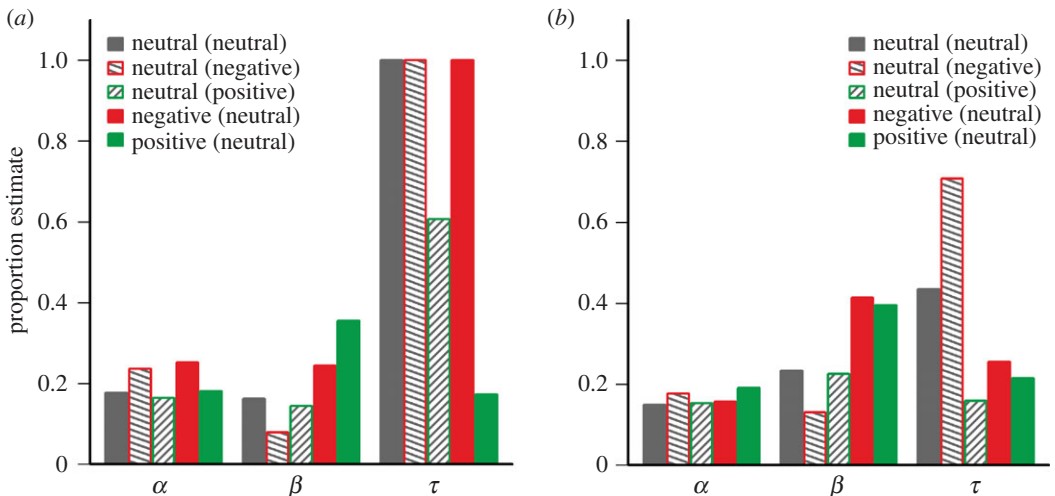

**Figure 4.** Parameter estimates of exponential decay functions fitted to the aggregated data in the first (a) and second (b) block of experiment 2, indicating the initial availability of stimulus information in iconic memory ($\alpha$), the amount of information that was transferred from iconic memory to visual short-term memory ($\beta$), and the degree of exponential decay of the initially available information ($\tau$; with higher values indicating less decay).

s.d. = 0.17, dog: $M = 0.34$; s.d. = 0.15, fish: $M = 0.28$; s.d. = 0.14, deer: $M = 0.17$; s.d. = 0.11), $F_{3,156} = 48.35$; $p < 0.001$; $\eta_G^2 = 0.32$.

### 4.2.2. Effects of distractor valence

It can also be seen in figure 3a,b that visual recognition of neutral targets was impaired when emotional distractors were presented (dashed lines). A 3 (distractor valence) × 5 (delay) repeated-measures ANOVA on recognition accuracy of neutral targets in the first block confirmed this main effect of distractor valence, $F_{2,104} = 6.68$; $p = 0.002$; $\eta_G^2 = 0.01$, suggesting that emotional distractors disrupt the selection of neutral information from iconic memory (positive distractors: $M = 0.25$; s.d. = 0.10, negative distractors: $M = 0.25$; s.d. = 0.10), as compared to neutral distractors ($M = 0.29$; s.d. = 0.10). Alpha-corrected pairwise comparisons revealed a significant contrast between negative and neutral distractors, $p = 0.005$, as well as between positive and neutral distractors, $p = 0.004$, but not between negative and positive distractors, $p = 0.69$. The analysis further revealed a significant main effect of delay, $F_{4,208} = 16.11$; $p < 0.001$; $\eta_G^2 = 0.07$, but no interaction, $F_{8,416} = 1.16$; $p = 0.32$; $\eta_G^2 < 0.01$, suggesting that the valence of non-target information in iconic memory did not affect the time course of the decay of neutral information in iconic memory with increasing cue delay.

These results were confirmed by a BF analysis base on the data in the first block of experiment 2, showing that a model 1 with two independent fixed effects of distractor valence and delay was more likely than a model 2 that also included the interaction term ($BF_{12} = 74.6$). Model 1 was also more likely than a null model with only delay as a fixed effect ($BF_{10} = 6.8$), whereas model 2 was not ($BF_{20} = 0.09$).

There was also an effect of distractor valence in the second block of experiment 2, $F_{2,104} = 13.68$; $p < 0.001$; $\eta_G^2 = 0.02$, with better recognition of neutral targets against neutral distractors ($M = 0.32$; s.d. = 0.13) as compared to negative ($M = 0.25$; s.d. = 0.11) or positive distractors ($M = 0.28$; s.d. = 0.11). Adjusted pairwise comparisons revealed significant contrasts between neutral and negative ($p < 0.001$), neutral and positive ($p = 0.011$), as well as between positive and negative distractors ($p = 0.02$), suggesting that negative distractors interfered more with the selection of neutral targets than positive distractors did (figure 3b). The ANOVA also revealed a significant main effect of delay in the second block, $F_{4,208} = 17.62$; $p < 0.001$; $\eta_G^2 = 0.07$, but no interaction, $F_{8,416} = 0.84$; $p = 0.57$; $\eta_G^2 < 0.01$. These results were again confirmed by the BF analysis, indicating that a model with two fixed effects of distractor valence and delay was more likely than a model that included the interaction term ($BF_{12} = 160.77$). Both models were more likely than the null model without distractor valence as a fixed effect ($BF_{10} = 1596.29$; $BF_{20} = 9.93$).

Statistical comparisons of distractor valence effects between the two blocks of experiment 2 revealed a marginally significant main effect of block, $F_{1,52} = 3.78$; $p = 0.06$; $\eta_G^2 < 0.01$, with only slight improvement

from the first ($M = 0.27$; s.d. = 0.08) to the second block ($M = 0.28$; s.d. = 0.10), and an also marginally significant interaction between distractor valence and block, $F_{2,104} = 2.48$; $p = 0.09$; $\eta^2_G < 0.01$, with slightly more pronounced effects of distractor valence in the second block (figure 3b). There were no other interactions with block, $F < 1.31$; $p > 0.27$.

Consistent with the effects of target valence, the exponential decay functions that were fitted to individual data (across both blocks) revealed a significant effect of distractor valence on the selection parameter $\beta$, $F_{2,104} = 7.73$; $p < 0.001$; $\eta^2_G = 0.08$, with higher estimates of $\beta$ for neutral targets that were to be selected against neutral distractors ($M = 0.21 \pm 0.02$), as compared to positive ($M = 0.18 \pm 0.01$) and negative distractors ($M = 0.14 \pm 0.02$). Again, there was no significant effect of distractor valence on the initial availability parameter $\alpha$, $F_{2,104} < 0.01$; $p > 0.99$; $\eta^2_G < 0.01$, and the temporal parameter $\tau$, $F_{2,104} = 1.36$; $p = 0.26$; $\eta^2_G = 0.02$.

### 4.2.3. Discussion

Experiment 2 successfully replicated the basic effects of emotional targets and distractors using a visual recognition task instead of having participants to verbally recall the labels of images selected from iconic memory. The data clearly show that both positive and negative images were recognized at higher accuracy compared to neutral images. In addition, the recognition of neutral targets against emotional distractors was further impaired relative to neutral distractors, indicating that the prioritization of emotional information in iconic memory interfered with the goal of retrieving a neutral target. This consistent pattern of results suggests that the response options shown during the recognition task did not interfere with the information held in visual working memory. Hence, the selection of information form iconic memory could be measured reliably using a non-verbal visual recognition task.

## 5. General discussion

In line with previous findings of prioritized sensory and cognitive processing of emotional information [5], the present two experiments demonstrated that emotional stimuli are more likely to be selected from iconic memory than neutral stimuli. Specifically, both verbal recall (experiment 1) and visual recognition accuracy (experiment 2) was higher for positive and negative (threatening) targets from a briefly presented set of stimuli than for neutral targets, regardless of the delay of the cue. In addition to this main effect of target valence, there was only some indication of these valence effects to depend on the delay of the retrieval cue. Only when the data were collapsed across both blocks—corresponding to the double number of trials compared to Kuhbandner et al. [11]—there was a significant interaction between target valence and delay on verbal recall in experiment 1, but not on visual recognition accuracy in experiment 2. This suggests that differences in the slopes of decay curves can be demonstrated with enhanced precision. However, the results of additional BF analyses revealed that it is most likely that the effect of target valence is independent of the effect of cue delay (i.e. there was very little evidence for an interaction model). Hence, the present data show that the emotionality of targets reliably affects the selection of information from iconic memory, but not necessarily the duration of information that was initially encoded in iconic memory. In addition, we were able to confirm the observation in the original study [11] of emotional distractors to interfere with the selection of neutral targets from iconic memory. Together, this successful replication and extension of a previously reported result [11] suggests that the emotional significance of a stimulus is processed already at a very early stage of visual perception.

Nevertheless, we also note that the emotional prioritization of negative and positive images in iconic memory was found to differ considerably between the particular exemplars used within each valence category. Specifically, the direct replication of the original revealed that images such as the spider, heart, or tree were much more likely to be recalled verbally than images such as the skull, butterfly or deer. This could possibly be explained with certain lexical entries being more accessible than others. However, a very similar exemplar bias was also observed in the extension experiment using a visual recognition task in which participants did not have to retrieve verbal labels from their mental lexicon. Hence, this finding suggests that the differences in iconic memory between exemplars of the same valence category may have been owing to geometric properties of certain images (e.g. the proportion of round or spiky shapes) which could make them visually more salient and thus easier to access from iconic memory than others. Considering the relatively small set of only 12 stimuli used in the present two experiments as well as in the original study, this observation might imply that the

findings are specific to the particular selection of stimuli rather than to the emotional meaning of the stimuli alone. We note that this might point to a severe limitation, which should be addressed in future studies using either different sets of stimuli or methods to account for factors affecting stimulus visibility (e.g. shape, frequency and similarity—as it has been investigated for letters, [27]) in order to rule out the possibility that the observed effects on iconic memory are driven by stimulus properties other than the emotional meaning.

Exponential decay functions were fitted to the accuracy data in order to describe the temporal dynamics of information processing in iconic memory. Specifically, these functions allow us to distinguish between the initial availability of information in iconic memory (parameter $\alpha$), the subsequent attentional selection of information for further processing (parameter $\beta$) and the extent of temporal decay of information that was initially available in iconic memory (parameter $\tau$). Consistent with Kuhbandner *et al.* [11], it was found with the verbal recall task in experiment 1 that emotional targets were more likely than neutral targets to be encoded initially in iconic memory (i.e. $\alpha$ was enhanced for positive and negative targets), whereas the same functions fitted to visual recognition accuracy in experiment 2 did not reveal any differences in initial availability as a function of target valence. This seems to indicate that verbal recall may indeed be a more sensitive method to demonstrate differences in terms of the initial encoding of information in iconic memory, whereas interference processes at the level of visual working memory during the visual recognition task could have eliminated these initial availability effects. However, emotional stimuli (and in particular the negative items) were found to be more likely to be selected for further processing in working memory regardless of the recall task (i.e. target valence significantly affected the attentional selection parameter $\beta$ in both experiments). This suggests that early sensory effects on the attentional selection of information for further processing in working memory can be detected with both verbal recall and the visual recognition measure used in the present extension experiment. Finally, in contrast to what has been reported previously [11], we were not able to demonstrate that the decay of information that was initially available in iconic memory was prolonged for negative targets compared to positive or neutral targets. Specifically, there was only a trend of prolonged availability of negative information in iconic memory in the second block of experiment 1 and in the first block of experiment 2, but we were not able to measure reliable effects of target valence on the temporal decay parameter $\tau$.

There was also very little evidence for a valence asymmetry with regard to the processing of emotional target information, because (i) negative and positive stimuli were equally likely to be selected from iconic memory for further processing (parameter $\beta$), and (ii) the decay of negative targets was only slightly longer than the decay of positive (and neutral) targets. Therefore, as also mentioned by Kuhbandner *et al.* [11], the finding of both positive and negative valence effects on iconic memory is consistent with the recent literature showing that reliable attentional biases can also be produced by positive emotional stimuli [33]. The findings are less consistent with evolutionary accounts proposing an early prioritization only for negative, fear-relevant stimuli serving adaptive purposes [34], but they seem to indicate that any stimulus that is potentially relevant to an individual's current goals may capture attention, regardless of its valence.

In addition to Kuhbandner *et al.* [11], who reported a non-significant trend for the effect of distractor valence, the present two experiments also revealed a reliable effect of emotional distractor images on the attentional selection of neutral targets from iconic memory. Specifically, neutral targets were less likely to be selected when emotional distractors were available in iconic memory than when only neutral information was present. This main effect of distractor valence was observed consistently in both task blocks with verbal recall in experiment 1 and with visual recognition in experiment 2, although the distractor interference effect appears to be more robust for negative distractors than for positive distractors (e.g. see the first block of experiment 1). The exponential decay curves further indicate that distractor valence affected the attentional selection parameter (significantly in experiment 2) more than the initial availability and the temporal decay parameters. This pattern of results suggests that the emotionality of information in iconic memory does not only facilitate processing of the to-be-selected information, but it appears to also disrupt the selection of other (neutral) information in iconic memory.

In the original study [11], it has been argued that only a limited number of trials could be performed by each participant in order to avoid attentional habituation effects with regard to the processing of emotional stimuli [21]. Both experiments of the present study involved two blocks each consisting of the original 200 trials allowing this assumption to be tested. While performance on both tasks (verbal recall and visual recognition) generally improved from the first to the second block, no evidence was found for the emotional effects on iconic memory to attenuate with increasing task practice. That is, both the facilitation observed for emotional targets as well as the interference produced by emotional

distractors were observed in both blocks of the task, indicating that the prioritization of emotional information in iconic memory was not susceptible to attentional habituation effects.

## 6. Conclusion

Taken together the results of the present replication project show that both positive and negative stimuli are more likely than neutral stimuli to be selected from iconic memory for further processing in working memory. This was observed with both a verbal recall and a visual recognition task, suggesting that the emotional effects on *attentional selection* are resistant to possible intrusions in visual short-term memory which might occur during visual recognition. Moreover, there is some indication of the emotional significance of a target to influence also its *initial availability* in iconic memory, at least if measured with immediate verbal recall (but not with visual recognition). By contrast, the emotional meaning does not seem to reliably influence the subsequent *decay* of information that was initially encoded in iconic memory. In addition to the prioritized attentional selection of emotional targets from iconic memory, the results of both experiments also demonstrated that emotional information which is available in iconic memory, but which is not to be selected, interferes with the selection of other neutral targets. This pattern of both target and distractor valence effects on the selection from iconic memory indicates that the emotional meaning of a stimulus must be processed at a very early stage of visual information processing, leading to either facilitation or disruption of attentional selection processes. It is important to note that the converging findings observed with verbal responses and visual recognition suggest that the prioritization of emotional information in iconic memory is a robust effect which may generalize across different experimental paradigms.

Ethics. The study was conducted strictly in accordance with the Declaration of Helsinki for experimentation with human participants (7th revision) and the APA Ethical Principles of Psychologists. The study was approved by the ethics committee of the Technical University of Darmstadt on 13 February 2017 (EK 04/2017). Participants of both experiments gave written informed consent before starting the tasks.

Data accessibility. The data of both experiments and the R code to conduct the statistical analysis and to reproduce the figures can be found on the OSF project page using this link: https://osf.io/kj69f/?view_only=ebe5fdbaf029465ab01bdac9ee2f445d.

Authors' contributions. Both authors developed the study concept, A.C. planned and organized the data collection, F.K. and A.C. performed the data analysis, F.K. drafted the manuscript and both authors provided critical revisions.

Competing interests. We declare we have no competing interests.

Funding. This work was supported by the German Research Foundation (DFG) with a grant to the first author (grant no. KA 4282/2-1).

Acknowledgements. We thank Lana Gerheim and Saskia Westerweller for their help with the data collection. We are also grateful to Christof Kuhbandner for sharing the original stimulus materials.

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
