## [Reviewer comments · Royal Society Open Science]

Review History

RSOS-190902.R0 (Original submission)

Review form: Reviewer 1

Do you have any ethical concerns with this paper?

No

Have you any concerns about statistical analyses in this paper?

Yes

Recommendation?

Major revision

Comments to the Author(s)

I appreciate the value of replications and think the paper addresses an interesting topic. There are, however, several problems with the proposal.

1. The proposed project is a conceptual replication, not a close replication. The authors aim to use a recognition test instead of a recall test. This a major change from the original study. In long-

term memory at least, findings can dramatically change depending on whether a recall or recognition task is used.

There may well be good reasons for using a recognition task, but I think this is a secondary issue. If the present study does not replicate the original study this might simple be due to the use of a different task. It would not indicate that the findings would not replicate in a close replication. If the authors think the use of recognition task is highly important or interesting they should run two experiments, one with verbal recall (a close replication of the original study), one with a visual recognition task (a conceptual replication).

I am not claiming that the use of a recognition task is uninteresting but I am not convinced that it eliminates contributions from phonological memory. True, the task may be performed solely on the basis of visual representations, but nothing prevents participants from using phonological codes.

2. To be honest, I thought that a power of .8 was somewhat meager. A power of .9 could be achieved with just 68 participants (given $d_z = 0.36$). Why not aim for higher statistical power?

3. The authors aim to remove participants whose recognition scores are not significantly above chance. Please provide details, about how this is calculated and what criterion is used. Will removed participants be replaced with new ones? If not, this will reduce the statistical power! Will the final report mention how many participants have been removed? Are pilot data available to indicate that participants are generally away from ceiling or floor?

4. The experiment manipulates valence of targets and distractors. Perhaps I misunderstood something but is the appropriate analysis not a 3 (target valence) x 3 (distractor valence) x 5 (delay) ANOVA?

5. The analyses are repeated separately for the first 200 and the second 200 trials. If the previous results are replicated in the first 200 trials, but not the second 200 trials, does this then count as a replication of the original findings? What if the reverse pattern is found? It seems to me we now have three shots at a replication: in the overall analysis, the first 200 trials and the second 200 trials. This does complicate interpretation of the results.

6. Please specify exactly how t-tests are corrected for multiple comparisons.

7. It was not clear to me whether data have already been collected or not. The participant section suggests participants have been tested already.

Review form: Reviewer 2 (Christof Kuhbandner)

Do you have any ethical concerns with this paper?

No

Have you any concerns about statistical analyses in this paper?

No

Recommendation?

Major revision

Comments to the Author(s)

Summary:

The reviewed paper by Kattner and Clausen is a Replication Study with the aim to replicate a the study by Kuhbandner, Spitzer, and Pekrun on the prioritization and decay of emotionally significant information in iconic memory, published in *Psychological Science* in 2011. I am the first author of that study (Kuhbandner). I appreciate it a lot that Kattner and Clausen have planned to replicate our study, and it is great to hear that our study is considered important enough by them to be replicated.

The paper seems to be a “results-blind track” submission – Stage 1, where the authors have already completed a replication attempt of a study. In fact, I have already reviewed a paper of the authors on this replication attempt, including data and analysis, which had been submitted to the journal *Psychological Science*; the paper was rejected by the editor since the paper had been submitted as a pre-registered replication report although the data had already been collected by the authors - which should not be a problem in case of the current submission because the paper seems to have been submitted as a “results-blind track” submission.

Below, I will reiterate my concerns in my review for *Psychological Science*. The paper provides a sufficiently clear and detailed description of the methods and analysis pipeline, of the logic, rationale, and plausibility of the proposed hypotheses, and the authors have considered sufficient outcome-neutral conditions for ensuring that the results obtained are able to test the stated hypotheses. However, in my opinion, the main problem is that the power of the replication study is too low, which may prevent the drawing of sufficiently valid and robust (e.g. statistically powerful) conclusions from the replication study. I will shortly summarize my concerns below.

Major Points

1) Sample Size

In the paper, the authors state that “a power analysis for the main effect of emotional significance of the immediate read-out of information from iconic memory at the shortest cue delay revealed a sample size of 50 participants to be required for an effect size of $d_z = 0.36$ ($M = .83$; $SEM = .075$ for emotional items vs. $M = .65$; $SEM = .075$, based on Fig. 1b in (11)) to be detected with a statistical power of $1 - \beta = .8$ and $\alpha = .05$ ” (p. 7, line 31).

Basically, as far as I know, the typical convention is that power is set higher in replication studies. For instance, in the recent replication projects such as the reproducibility project by Brian Nosek and colleagues, power was set to 0.95. Furthermore, in our study, we did not run t-tests but likelihood-ratio tests because our analyses were performed using the aggregate data rather than individually estimated parameters (see footnote 1 in our paper). Thus, the reported power analysis may not be adequate.

Furthermore, the authors argue that one of their main aim was to clarify null effects in our original study (e.g., no difference in speed of decay between neutral and positive stimuli). When the aim is to clarify null effects (i.e., non-significant effects), one has to take into account the effect size of the to-be-replicated null effect, that is in the current context, the effect of positive emotional significance on decay (and not simply the main effect of emotional significance at the shortest cue delay). Since the observed effect size in the to-be-replicated study was relatively small ($\chi^2(2) < 1$, $p > .60$), a much larger sample size would be needed to definitely clarify whether the null-effect observed for positive stimuli was a false-negative. The problem can also be made aware in a more intuitive way: If one tries to show with a new study that a non-significant effect of an original study is actually significant, it does not make much sense to run a new study with about the same sample size, and then to compare significance levels. To rule out whether the effect actually is there in such a situation, the only informative way is either to run a new experiment with a substantially larger sample size, or to combine the data across the two experiments with meta-analytical techniques.

The same holds true for the aim of the authors to clarify possible interference effects of positive and threatening non-target stimuli (distractors) in iconic memory. This was also part of our study, where we have also manipulated the emotional significance of distractors that were presented together with neutral targets, and where we have observed a clear tendency for lower performance in the conditions with emotional distractors (see Fig. 1b). However, the sample size of the original study (45 participants) was too small to reveal significant effects. The attempt to clarify interference effects by emotional distractors is indeed an important question. However, to definitely clarify this question, a substantially larger sample size than that used in the submitted replication study is necessary given that the effects seem to be rather small.

2) Methodological changes between the original and the replication study

In their replication attempt, the authors have replaced the verbal free recall test in the iconic memory tests of the original study by a visual 12-alternative forced-choice recognition task where the possible twelve pictures are available at the bottom of the screen for the participant's response. However, by presenting all 12 possible stimuli visually in a recognition test, interference and time delay effects may be introduced that may cloud true memory abilities. On each trial, participants are asked about the specific picture that was presented at the cued position. When using a 12-alternative choice task, participants have to go through all 12 possible pictures which may introduce (1) interference by pictures that were not the targets, and (2) an additional time delay because it may take some time until one reaches at the correct answer. Furthermore (3), participants have to change their visual focus on the screen to provide an answer, which may possibly introduce additional problematic effects. In their submitted paper, the authors already try to argue against these potential problems (p. 6, line 29 ff) because these concerns have already been raised in the former reviews they have received from Psychological Science. I definitely leave it open to the editor to evaluate the (non)severity of these changes in methods between the original and the replication study.

Decision letter (RSOS-190902.R0)

09-Jul-2019

Dear Dr Kattner,

The Editors assigned to your Stage 1 Replication submission ("Revisiting the prioritization of emotional information in iconic memory") have now received comments from reviewers. We would like you to revise your paper in accordance with the referee and editors suggestions which can be found below (not including confidential reports to the Editor). Please note this decision does not guarantee eventual acceptance.

When submitting your revised manuscript, you must respond to the comments made by the referees and upload a file "Response to Referees" in the "File Upload" step. Please use this to document how you have responded to the comments, and the adjustments you have made. In order to expedite the processing of the revised manuscript, please be as specific as possible in your response.

Once again, thank you for submitting your manuscript to Royal Society Open Science and I look

forward to receiving your revision. If you have any questions at all, please do not hesitate to get in touch. Full author guidelines may be found at <http://rsos.royalsocietypublishing.org/page/replication-studies#AuthorsGuidance>.

Kind regards,
Professor Chris Chambers
Registered Reports Editor
Royal Society Open Science
openscience@royalsociety.org

Editor Comments to Author (Professor Chris Chambers):

Two expert reviewers have now assessed the manuscript. Both reviewers find that either one or both of the Stage 1 primary criteria are not yet met. In particular, the reviewers note two considerable weaknesses with the submission: suboptimal statistical power, and (more seriously for this format) major deviations from the original target study (including replacement of the recall task with a recognition task). Meeting both Stage 1 primary criteria is essential for achieving in-principle acceptance as a Replication, and for this reason, the manuscript falls very close to the line for an outright Stage 1 rejection.

I would, however, like to invite the authors to submit a Major Revision in which both concerns -- and especially the deviation from the original study -- are convincingly addressed. Based on the authors' response (and possible re-review), I will then decide either to issue IPA or to reject the submission. In case of a rejection, one possible outcome is that the authors will be invited to perform an additional higher powered, closer replication of the original study and that the indirect and direct replications could be published together in the same paper.

Comments to Author:

Reviewer: 1

I appreciate the value of replications and think the paper addresses an interesting topic. There are, however, several problems with the proposal.

1. The proposed project is a conceptual replication, not a close replication. The authors aim to use a recognition test instead of a recall test. This is a major change from the original study. In long-term memory at least, findings can dramatically change depending on whether a recall or recognition task is used.

There may well be good reasons for using a recognition task, but I think this is a secondary issue. If the present study does not replicate the original study this might simply be due to the use of a different task. It would not indicate that the findings would not replicate in a close replication. If the authors think the use of recognition task is highly important or interesting they should run two experiments, one with verbal recall (a close replication of the original study), one with a visual recognition task (a conceptual replication).

I am not claiming that the use of a recognition task is uninteresting but I am not convinced that it eliminates contributions from phonological memory. True, the task may be performed solely on the basis of visual representations, but nothing prevents participants from using phonological codes.

2. To be honest, I thought that a power of .8 was somewhat meager. A power of .9 could be achieved with just 68 participants (given $d_z = 0.36$). Why not aim for higher statistical power?

3. The authors aim to remove participants whose recognition scores are not significantly above chance. Please provide details, about how this is calculated and what criterion is used. Will removed participants be replaced with new ones? If not, this will reduce the statistical power!

Will the final report mention how many participants have been removed? Are pilot data available to indicate that participants are generally away from ceiling or floor?

4. The experiment manipulates valence of targets and distractors. Perhaps I misunderstood something but is the appropriate analysis not a 3 (target valence) x 3 (distractor valence) x 5 (delay) ANOVA?

5. The analyses are repeated separately for the first 200 and the second 200 trials. If the previous results are replicated in the first 200 trials, but not the second 200 trials, does this then count as a replication of the original findings? What if the reverse pattern is found? It seems to me we now have three shots at a replication: in the overall analysis, the first 200 trials and the second 200 trials. This does complicate interpretation of the results.

6. Please specify exactly how t-tests are corrected for multiple comparisons.

7. It was not clear to me whether data have already been collected or not. The participant section suggests participants have been tested already.

Reviewer: 2

Comments to the Author(s)

Summary:

The reviewed paper by Kattner and Clausen is a Replication Study with the aim to replicate a the study by Kuhbandner, Spitzer, and Pekrun on the prioritization and decay of emotionally significant information in iconic memory, published in *Psychological Science* in 2011. I am the first author of that study (Kuhbandner). I appreciate it a lot that Kattner and Clausen have planned to replicate our study, and it is great to hear that our study is considered important enough by them to be replicated.

The paper seems to be a “results-blind track” submission – Stage 1, where the authors have already completed a replication attempt of a study. In fact, I have already reviewed a paper of the authors on this replication attempt, including data and analysis, which had been submitted to the journal *Psychological Science*; the paper was rejected by the editor since the paper had been submitted as a pre-registered replication report although the data had already been collected by the authors - which should not be a problem in case of the current submission because the paper seems to have been submitted as a “results-blind track” submission.

Below, I will reiterate my concerns in my review for *Psychological Science*. The paper provides a sufficiently clear and detailed description of the methods and analysis pipeline, of the logic, rationale, and plausibility of the proposed hypotheses, and the authors have considered sufficient outcome-neutral conditions for ensuring that the results obtained are able to test the stated hypotheses. However, in my opinion, the main problem is that the power of the replication study is too low, which may prevent the drawing of sufficiently valid and robust (e.g. statistically powerful) conclusions from the replication study. I will shortly summarize my concerns below.

Major Points

1) Sample Size

In the paper, the authors state that “a power analysis for the main effect of emotional significance of the immediate read-out of information from iconic memory at the shortest cue delay revealed a sample size of 50 participants to be required for an effect size of $d_z = 0.36$ ($M = .83$; $SEM = .075$ for emotional items vs. $M = .65$; $SEM = .075$, based on Fig. 1b in (11)) to be detected with a statistical power of $1 - \beta = .8$ and $\alpha = .05$ ” (p. 7, line 31).

Basically, as far as I know, the typical convention is that power is set higher in replication studies.

For instance, in the recent replication projects such as the reproducibility project by Brian Nosek and colleagues, power was set to 0.95. Furthermore, in our study, we did not run t-tests but likelihood-ratio tests because our analyses were performed using the aggregate data rather than individually estimated parameters (see footnote 1 in our paper). Thus, the reported power analysis may not be adequate.

Furthermore, the authors argue that one of their main aim was to clarify null effects in our original study (e.g., no difference in speed of decay between neutral and positive stimuli). When the aim is to clarify null effects (i.e., non-significant effects), one has to take into account the effect size of the to-be-replicated null effect, that is in the current context, the effect of positive emotional significance on decay (and not simply the main effect of emotional significance at the shortest cue delay). Since the observed effect size in the to-be-replicated study was relatively small ($\chi^2(2) < 1, p > .60$), a much larger sample size would be needed to definitely clarify whether the null-effect observed for positive stimuli was a false-negative. The problem can also be made aware in a more intuitive way: If one tries to show with a new study that a non-significant effect of an original study is actually significant, it does not make much sense to run a new study with about the same sample size, and then to compare significance levels. To rule out whether the effect actually is there in such a situation, the only informative way is either to run a new experiment with a substantially larger sample size, or to combine the data across the two experiments with meta-analytical techniques.

The same holds true for the aim of the authors to clarify possible interference effects of positive and threatening non-target stimuli (distractors) in iconic memory. This was also part of our study, where we have also manipulated the emotional significance of distractors that were presented together with neutral targets, and where we have observed a clear tendency for lower performance in the conditions with emotional distractors (see Fig. 1b). However, the sample size of the original study (45 participants) was too small to reveal significant effects. The attempt to clarify interference effects by emotional distractors is indeed an important question. However, to definitely clarify this question, a substantially larger sample size than that used in the submitted replication study is necessary given that the effects seem to be rather small.

2) Methodological changes between the original and the replication study

In their replication attempt, the authors have replaced the verbal free recall test in the iconic memory tests of the original study by a visual 12-alternative forced-choice recognition task where the possible twelve pictures are available at the bottom of the screen for the participant's response. However, by presenting all 12 possible stimuli visually in a recognition test, interference and time delay effects may be introduced that may cloud true memory abilities. On each trial, participants are asked about the specific picture that was presented at the cued position. When using a 12-alternative choice task, participants have to go through all 12 possible pictures which may introduce (1) interference by pictures that were not the targets, and (2) an additional time delay because it may take some time until one reaches at the correct answer. Furthermore (3), participants have to change their visual focus on the screen to provide an answer, which may possibly introduce additional problematic effects. In their submitted paper, the authors already try to argue against these potential problems (p. 6, line 29 ff) because these concerns have already been raised in the former reviews they have received from Psychological Science. I definitely leave it open to the editor to evaluate the (non)severity of these changes in methods between the original and the replication study.

Author's Response to Decision Letter for (RSOS-190902.R0)

See Appendix A.

RSOS-190902.R1 (Revision)

Review form: Reviewer 1

Do you have any ethical concerns with this paper?

No

Have you any concerns about statistical analyses in this paper?

No

Recommendation?

Major revision

Comments to the Author(s)

The authors have satisfactorily addressed most of my comments.

I am still not convinced, however, that the change from verbal recall to recognition is appropriate. We have not seen the results and have to consider two possible outcomes. Either the results of the original study are replicated or they are not. If they are, I think most readers will be happy to accept that those of the original study and of the present study are true (reflect differences in the population). If, however, the results of the original study do not replicate I think most readers will wonder whether the original findings do not generalize to the new task or whether they do no replicate at all (not even if the same task is used as in the original study). We have to seriously consider the possibility that the original study does not replicate (if we don't, why bother publishing the present study?) and I think it would be to the better to include two experiments: one using verbal recall and the experiment using a recognition task.

Decision letter (RSOS-190902.R1)

23-Aug-2019

Dear Dr Kattner,

The Editors assigned to your Stage 1 Replication submission ("Revisiting the prioritization of emotional information in iconic memory") have now received comments from reviewers. We would like you to revise your paper in accordance with the referee and editors suggestions which can be found below (not including confidential reports to the Editor). Please note this decision does not guarantee eventual acceptance.

Once again, thank you for submitting your manuscript to Royal Society Open Science and I look forward to receiving your revision. If you have any questions at all, please do not hesitate to get in touch. Full guidance for authors is also available at <http://rsos.royalsocietypublishing.org/page/replication-studies#AuthorsGuidance>.

on behalf of Professor Chris Chambers (Registered Reports Editor, Royal Society Open Science)
openscience@royalsociety.org

Associate Editor Comments to Author (Professor Chris Chambers):

The revised manuscript was returned to one of the previous reviewers for reconsideration. The reviewer judges the manuscript to be much improved but remains unsatisfied with the deviation from verbal recall to recognition, and on this basis judges that Stage 1 Primary Criterion 2 is not met. This issue appears straightforward (albeit laborious) to address by conducting a closer replication using verbal recall and reporting both experiments in the final paper. If the authors are able to conduct this additional experiment then we would be happy to consider a revised Stage 1 submission that includes both study designs, including the direct and indirect replications.

Comments to Author:

Reviewer: 1

The authors have satisfactorily addressed most of my comments.

I am still not convinced, however, that the change from verbal recall to recognition is appropriate. We have not seen the results and have to consider two possible outcomes. Either the results of the original study are replicated or they are not. If they are, I think most readers will be happy to accept that those of the original study and of the present study are true (reflect differences in the population). If, however, the results of the original study do not replicate I think most readers will wonder whether the original findings do not generalize to the new task or whether they do not replicate at all (not even if the same task is used as in the original study). We have to seriously consider the possibility that the original study does not replicate (if we don't, why bother publishing the present study?) and I think it would be to the better to include two experiments: one using verbal recall and the experiment using a recognition task.

Author's Response to Decision Letter for (RSOS-190902.R1)

See Appendix B.

Decision letter (RSOS-191507.R0)

10-Sep-2019

Dear Dr Kattner

On behalf of the Editor, I am pleased to inform you that your Stage 1 Replication, RSOS-191507

entitled "Revisiting the prioritization of emotional information in iconic memory" has been accepted in principle for publication in Royal Society Open Science. The reviewers' and editors' comments are included at the end of this email.

Please note that you must now register your approved protocol on the Open Science Framework (<https://osf.io/rr>), using the 'Submit your approved Registered Report' option and then the 'Registered Report Protocol Preregistration' option. Please use the Registered Report option even though your article is being accepted as a Stage 1 Replication.

Further into the registration process, in the Journal Title field enter 'Royal Society Open Science (Replication article type)'. Please note that a time-stamped, independent registration of the protocol is mandatory under journal policy, and manuscripts that do not conform to this requirement cannot be considered at Stage 2. The protocol should be registered unchanged from its current approved state. Please include a URL to the protocol in your Stage 2 manuscript, and because you submitted via the Results-Blind track (but also, after review, added a protocol for an experiment yet to be completed) please note in the manuscript that some of the preregistration was performed after data analysis (e.g. 'This article received in-principle acceptance (IPA) at Royal Society Open Science on [insert date]. Following IPA, the accepted Stage 1 version of the manuscript, not including results and discussion, was preregistered on the OSF [insert URL]. The preregistration for Experiment X was undertaken after data analysis; the preregistration for Experiment Y was undertaken before data analysis.')

Following completion of your study, we invite you to resubmit your paper for peer review as a Stage 2 Replication. Please note that your manuscript can still be rejected for publication at Stage 2 if the Editors consider any of the following conditions to be met:

- The Introduction and methods deviated from the approved Stage 1 submission (required).
- The authors' conclusions were not considered justified given the data.

We encourage you to read the complete guidelines for authors concerning Stage 2 submissions at : <http://rsos.royalsocietypublishing.org/page/replication-studies#AuthorsGuidance>. Please especially note the requirements for data sharing and that withdrawing your manuscript will result in publication of a Withdrawn Registration.

We encourage you to read the complete guidelines for authors concerning Stage 2 submissions at <https://royalsocietypublishing.org/rsos/registered-reports#ReviewerGuideRegRep>. Please especially note the requirements for data sharing and that withdrawing your manuscript will result in publication of a Withdrawn Registration.

Once again, thank you for submitting your manuscript to Royal Society Open Science and I look forward to receiving your Stage 2 submission. If you have any questions at all, please do not hesitate to get in touch. We look forward to hearing from you shortly with the anticipated submission date for your stage two manuscript.

Kind regards,
Professor Chris Chambers
Royal Society Open Science
openscience@royalsociety.org

Author's Response to Decision Letter for (RSOS-191507.R0)

See Appendix C.

RSOS-191507.R1 (Revision)

Review form: Reviewer 1

Do you have any ethical concerns with this paper?

No

Have you any concerns about statistical analyses in this paper?

No

Recommendation?

Accept with minor revision

Comments to the Author(s)

The addition of Experiment 1 (verbal recall), a direct replication of the Kuhbandner et al., 2011 (Psychological Science, 22, 695-700) study, was essential. I think the present paper is (almost) publishable.

1. Were the 12 drawings (page 9, Apparatus and Stimuli) identical to those of the original study?
2. The differences in performance for different stimuli within each of the three categories (positive, negative, neutral) was rather large. This is acknowledged and discussed on page 30. Given the small set of stimuli 12 (4 positive, 4 negative, 4 neutral) this means that the current findings are particularly sensitive to the stimuli that happened to be included in the experiment. Different results might have been obtained with a different selection of stimuli. I am not suggesting that the authors run another replication study with a new stimulus set before the results are published but a more explicit statement about this limitation could be included in the discussion.

Decision letter (RSOS-191507.R1)

Dear Dr Kattner

On behalf of the Editor, I am pleased to inform you that your Stage 2 Replication submission RSOS-191507.R1 entitled "Revisiting the prioritization of emotional information in iconic memory" has been accepted for publication in Royal Society Open Science subject to minor revision in accordance with the referee suggestions. Please find the referees' comments at the end of this email.

The reviewers and Subject Editor have recommended publication, but also suggest some minor revisions to your manuscript. Therefore, I invite you to respond to the comments and revise your manuscript.

Please also ensure that all the below editorial sections are included where appropriate (a non-exhaustive example is included in an attachment):

- Ethics statement

- Data accessibility

<http://datadryad.org/submit?journalID=RSOS&manu=RSOS-191507.R1>

- Competing interests

- Authors' contributions

- Acknowledgements

- Funding statement

Because the schedule for publication is very tight, it is a condition of publication that you submit the revised version of your manuscript within 7 days (i.e. by the 17-Sep-2020). If you do not think you will be able to meet this date please let me know immediately.

To revise your manuscript, log into <https://mc.manuscriptcentral.com/rsos> and enter your Author Centre, where you will find your manuscript title listed under "Manuscripts with Decisions". Under "Actions," click on "Create a Revision." You will be unable to make your

revisions on the originally submitted version of the manuscript. Instead, revise your manuscript and upload a new version through your Author Centre.

- 1) A text file of the manuscript (tex, txt, rtf, docx or doc), references, tables (including captions) and figure captions. Do not upload a PDF as your "Main Document".
- 2) A separate electronic file of each figure (EPS or print-quality PDF preferred (either format should be produced directly from original creation package), or original software format)
- 3) Included a 100 word media summary of your paper when requested at submission. Please ensure you have entered correct contact details (email, institution and telephone) in your user account
- 4) Included the raw data to support the claims made in your paper. You can either include your data as electronic supplementary material or upload to a repository and include the relevant DOI within your manuscript
- 5) Included your supplementary files in a format you are happy with (no line numbers, Vancouver referencing, track changes removed etc) as these files will NOT be edited in production

Kind regards,
 Professor Chris Chambers
 Royal Society Open Science
 openscience@royalsociety.org

on behalf of Professor Chris Chambers (Associate Editor) and Chris Chambers (Registered Reports Editor, Royal Society Open Science)
 openscience@royalsociety.org

Associate Editor Comments to Author (Professor Chris Chambers):
 Comments to the Author:

One of the original Stage 1 reviewers was available to assess the Stage 2 manuscript, and I also read it myself. The reviewer's assessment is positive, with some minor suggestions for revising the discussion and clarifying methodological details. In addition, please add .csv versions of the data files to the OSF archive so that they are easily accessible to non-R users. Provided the authors are able to respond thoroughly to all points raised, final acceptance should be forthcoming without requiring further in-depth review.

Reviewers' comments to Author:
 Reviewer: 1

Comments to the Author(s)

The addition of Experiment 1 (verbal recall), a direct replication of the Kuhbandner et al., 2011 (Psychological Science, 22, 695-700) study, was essential. I think the present paper is (almost) publishable.

1. Were the 12 drawings (page 9, Apparatus and Stimuli) identical to those of the original study?
2. The differences in performance for different stimuli within each of the three categories (positive, negative, neutral) was rather large. This is acknowledged and discussed on page 30. Given the small set of stimuli 12 (4 positive, 4 negative, 4 neutral) this means that the current findings are particularly sensitive to the stimuli that happened to be included in the experiment. Different results might have been obtained with a different selection of stimuli. I am not suggesting that the authors run another replication study with a new stimulus set before the results are published but a more explicit statement about this limitation could be included in the discussion.

Author's Response to Decision Letter for (RSOS-191507.R1)

See Appendix D.

Decision letter (RSOS-191507.R2)

Dear Dr Kattner:

It is a pleasure to accept your Stage 2 Replication entitled "Revisiting the prioritization of emotional information in iconic memory" in its current form for publication in Royal Society Open Science.

on behalf of Professor Chris Chambers (Subject Editor)
openscience@royalsociety.org

Appendix A

Reviewer: 1

I appreciate the value of replications and think the paper addresses an interesting topic. There are, however, several problems with the proposal.

1. The proposed project is a conceptual replication, not a close replication. The authors aim to use a recognition test instead of a recall test. This a major change from the original study. In long-term memory at least, findings can dramatically change depending on whether a recall or recognition task is used.

There may well be good reasons for using a recognition task, but I think this is a secondary issue. If the present study does not replicate the original study this might simple be due to the use of a different task. It would not indicate that the findings would not replicate in a close replication. If the authors think the use of recognition task is highly important or interesting they should run two experiments, one with verbal recall (a close replication of the original study), one with a visual recognition task (a conceptual replication).

I am not claiming that the use of a recognition task is uninteresting but I am not convinced that it eliminates contributions from phonological memory. True, the task may be performed solely on the basis of visual representations, but nothing prevents participants from using phonological codes.

We thank the reviewer for this important comment and we agree that recall and recognition response procedures may produce very different results for long-term memory measures. However, we do not think that this is the case when testing iconic memory because the selection of information from iconic memory is expected to occur very rapidly and is likely to be completed when participants are required to visually recognize the target stimulus. More specifically, we argue that the information that was retrieved from iconic memory within the first 500 ms or so (while participants fixate the central fixation cross) will already be stored in working memory when participants make their first eye movements from the central fixation to the response options shown at the bottom of the screen. So while the visual recognition task requires participants to match the visual items with information held in working memory, it is unlikely that this will affect the amount of information that was selected from iconic memory.

In contrast, verbal recall may require slightly more immediate retrieval of information from short-term memory, but it may also be affected by unwanted processes such as the interference between possible verbal labels for the visual information held in working memory and the preparation of articulatory processes. In a visual recognition task, it is most likely that participants use the information selected from iconic memory (held in working memory) as a template which can be matched with the response options shown at the bottom of the screen. This has been found to be a very fast and efficient process in visual search tasks (Malcolm & Henderson, 2009; Vickery, King, & Jiang, 2005). We have elaborated this point in the introduction of the revised manuscript on pp. 6-7.

Malcolm, G. L., & Henderson, J. M. (2009). The effects of target template specificity on visual search in real-world scenes: Evidence from eye movements. *Journal of Vision*, 9(11):8, 1-13. doi:10.1167/9.11.8

Vickery, T. J., King, L., & Jiang, Y. (2005). Setting up the target template in visual search. *Journal of Vision*, 5(1):8, 81-92. doi:10.1167/5.1.8

2. To be honest, I thought that a power of .8 was somewhat meager. A power of .9 could be achieved with just 68 participants (given $d_z = 0.36$). Why not aim for higher statistical power?

We totally agree with the reviewer, and we decided to increase the sample size to 68 participants in order to reach an enhanced statistical power of $1-\beta = .9$ (see p. 8 of the revised manuscript).

3. The authors aim to remove participants whose recognition scores are not significantly above chance. Please provide details, about how this is calculated and what criterion is used. Will removed participants be replaced with new ones? If not, this will reduce the statistical power! Will the final report mention how many participants have been removed? Are pilot data available to indicate that participants are generally away from ceiling or floor?

We thank the reviewer for these comments. We tested individual accuracy of responses against chance level ($p = .083$) using one-sample t tests. The data of participants were not included for the analyses if performance did not differ significantly from chance ($p > .05$). We have elaborated this information on p. 11, and the number of removed participants will be reported in the final manuscript. The removed participants will be replaced by new ones in order to reach a statistical power of $1-\beta = .9$. A pilot study was not required as we planned the experiment to be a very close replication of the original study, and we expected similar levels of accuracy.

4. The experiment manipulates valence of targets and distractors. Perhaps I misunderstood something but is the appropriate analysis not a 3 (target valence) x 3 (distractor valence) x 5 (delay) ANOVA?

We used the same experimental design as in the original study which did not include all factorial combinations of target valence and distractor valence. Specifically, distractor valence was manipulated only for the neutral target condition, and target valence was manipulated only for the neutral distractor condition. Accordingly, it is not possible to conduct a $3 \times 3 \times 5$ ANOVA, and we tested the effects of target valence and distractor valence separately.

5. The analyses are repeated separately for the first 200 and the second 200 trials. If the previous results are replicated in the first 200 trials, but not the second 200 trials, does this then count as a replication of the original findings? What if the reverse pattern is found? It seems to me we now have three shots at a replication: in the overall analysis, the first 200 trials and the second 200 trials. This does complicate interpretation of the results.

We thank the reviewer for this important comment. Only the first 200 trials are considered for the replication of the original study. The last 200 trials and the overall analysis are seen as an extension in order to (a) test for the emotional effects on iconic memory with enhanced statistical power (i.e., based on 400 trials), and (b) assess whether the emotional biases in iconic memory decrease over time (i.e., by comparing the first with the second 200 trials). We have elaborated this rationale in the manuscript (pp. 7-8).

6. Please specify exactly how t-tests are corrected for multiple comparisons.

The *t* tests were corrected by controlling for the false-discovery rate, using the method suggested by Benjamini and Hochberg (1995). This information is mentioned in the manuscript on p. 11.

Benjamini, Y., & Hochberg, Y. (1995). Controlling the false discovery rate: a practical and powerful approach to multiple testing. *Journal of the Royal Statistical Society Series B*, 57, 289–300.
<http://www.jstor.org/stable/2346101>.

7. It was not clear to me whether data have already been collected or not. The participant section suggests participants have been tested already.

Please note that we submitted a partial manuscript containing abstract, introduction and method section using the “results-blind track”. In line with the guidelines for authors (see below for the respective paragraph), we mentioned in the cover letter (to the editor) whether the data have already been collected or not, and we withheld this information for the stage-1 review process.

Excerpt of the guidelines for authors: “Note that while the cover letter should indicate whether results are known to the authors, the Stage 1 manuscript itself must not state or imply that the authors already know the results. To minimise potential bias against anticipated negative findings, reviewers in the first round of review will be blinded both to the results and to the existence of results.”

Reviewer: 2

Summary:

The reviewed paper by Kattner and Clausen is a Replication Study with the aim to replicate a the study by Kuhbandner, Spitzer, and Pekrun on the prioritization and decay of emotionally significant information in iconic memory, published in Psychological Science in 2011. I am the first author of that study (Kuhbandner). I appreciate it a lot that Kattner and Clausen have planned to replicate our study, and it is great to hear that our study is considered important enough by them to be replicated.

The paper seems to be a “results-blind track” submission – Stage 1, where the authors have already completed a replication attempt of a study. In fact, I have already reviewed a paper of the authors on this replication attempt, including data and analysis, which had been submitted to the journal Psychological Science; the paper was rejected by the editor since the paper had been submitted as a pre-registered replication report although the data had already been collected by the authors - which should not be a problem in case of the current submission because the paper seems to have been submitted as a “results-blind track” submission.

Below, I will reiterate my concerns in my review for Psychological Science. The paper provides a sufficiently clear and detailed description of the methods and analysis pipeline, of the logic, rationale, and plausibility of the proposed hypotheses, and the authors have considered sufficient outcome-neutral conditions for ensuring that the results obtained are

able to test the stated hypotheses. However, in my opinion, the main problem is that the power of the replication study is too low, which may prevent the drawing of sufficiently valid and robust (e.g. statistically powerful) conclusions from the replication study. I will shortly summarize my concerns below.

Major Points

1) Sample Size

In the paper, the authors state that “a power analysis for the main effect of emotional significance of the immediate read-out of information from iconic memory at the shortest cue delay revealed a sample size of 50 participants to be required for an effect size of $d_z = 0.36$ ($M = .83$; $SEM = .075$ for emotional items vs. $M = .65$; $SEM = .075$, based on Fig. 1b in (11)) to be detected with a statistical power of $1 - \beta = .8$ and $\alpha = .05$ ” (p. 7, line 31).

Basically, as far as I know, the typical convention is that power is set higher in replication studies. For instance, in the recent replication projects such as the reproducibility project by Brian Nosek and colleagues, power was set to 0.95. Furthermore, in our study, we did not run *t*-tests but likelihood-ratio tests because our analyses were performed using the aggregate data rather than individually estimated parameters (see footnote 1 in our paper). Thus, the reported power analysis may not be adequate.

Furthermore, the authors argue that one of their main aim was to clarify null effects in our original study (e.g., no difference in speed of decay between neutral and positive stimuli). When the aim is to clarify null effects (i.e., non-significant effects), one has to take into account the effect size of the to-be-replicated null effect, that is in the current context, the effect of positive emotional significance on decay (and not simply the main effect of emotional significance at the shortest cue delay). Since the observed effect size in the to-be-replicated study was relatively small ($\chi^2(2) < 1$, $p > .60$), a much larger sample size would be needed to definitely clarify whether the null-effect observed for positive stimuli was a false-negative. The problem can also be made aware in a more intuitive way: If one tries to show with a new study that a non-significant effect of an original study is actually significant, it does not make much sense to run a new study with about the same sample size, and then to compare significance levels. To rule out whether the effect actually is there in such a situation, the only informative way is either to run a new experiment with a substantially larger sample size, or to combine the data across the two experiments with meta-analytical techniques.

The same holds true for the aim of the authors to clarify possible interference effects of positive and threatening non-target stimuli (distractors) in iconic memory. This was also part of our study, where we have also manipulated the emotional significance of distractors that were presented together with neutral targets, and where we have observed a clear tendency for lower performance in the conditions with emotional distractors (see Fig. 1b). However, the sample size of the original study (45 participants) was too small to reveal significant effects. The attempt to clarify interference effects by emotional distractors is indeed an important question. However, to definitely clarify this question, a substantially larger sample size than that used in the submitted replication study is necessary given that the effects seem to be rather small.

We thank the reviewer for these important comments, and we agree that a replication study should generally be conducted with higher statistical power than the original study. We therefore decided to enhance the sample size to $N = 68$ in order to reach a power of .9 (see p. 8 of the manuscript and our response to Reviewer #1's point 2).

2) Methodological changes between the original and the replication study

In their replication attempt, the authors have replaced the verbal free recall test in the iconic memory tests of the original study by a visual 12-alternative forced-choice recognition task where the possible twelve pictures are available at the bottom of the screen for the participant's response. However, by presenting all 12 possible stimuli visually in a recognition test, interference and time delay effects may be introduced that may cloud true memory abilities. On each trial, participants are asked about the specific picture that was presented at the cued position. When using a 12-alternative choice task, participants have to go through all 12 possible pictures which may introduce (1) interference by pictures that were not the targets, and (2) an additional time delay because it may take some time until one reaches at the correct answer. Furthermore (3), participants have to change their visual focus on the screen to provide an answer, which may possibly introduce additional problematic effects. In their submitted paper, the authors already try to argue against these potential problems (p. 6, line 29 ff) because these concerns have already been raised in the former reviews they have received from Psychological Science. I definitely leave it open to the editor to evaluate the (non)severity of these changes in methods between the original and the replication study.

We thank the reviewer for these comments on the visual recognition task. In principle, we agree with all three points: A visual recognition task may induce visual interference, it requires additional time, and participants have to make an eye movement prior to making a response. However, as mentioned above, we argue that these effects are most likely to take place in visual working memory. Specifically, participants need to have transferred information from iconic memory to working memory in order to provide a response. The selection of information from iconic memory is most likely to occur much earlier while participants fixate the central fixation point. The target pictures are thus represented in the periphery of the iconic-memory image at equal eccentricity from the fovea, and participants need to select only one picture (based on the cue) and transfer it to working memory for later retrieval. Participants thus have to maintain only a single item in visual working memory, and we believe that it is highly unlikely that this item will be lost due to visual interference, the delay or the additional eye movement. Hence, accuracy on this task is expected to reflect the amount of information that was initially selected from iconic memory and transferred to working memory. Moreover, to make the visual recognition task as easy as possible for the participants, the twelve pictures were shown in the same order throughout the experiment (and randomised between participants). Hence, participants will learn the image locations, and they will have to go through much less than all 12 pictures on the majority of trials. That is, we do not think that the use of a recognition task should dramatically change observed results. Just as verbal recall, visual recognition of a single target item encoded in working memory should thus be sensitive to the present manipulations of iconic memory (i.e., delay and emotional meaning), and we consider the use of a recognition task an important minor modification to demonstrate generalisability of the effects observed in the original study.

Appendix B

To:
Prof. Chris Chambers
Cardiff University, UK
Associate Editor, *Royal Society Open Science*

29th August, 2019

Dear Professor Chambers,

We thank you for the opportunity to submit another revision of our manuscript entitled “Revisiting the prioritization of emotional information in iconic memory” which we previously submitted as a Stage-1 submission through the results-blind track for replication studies. As you will see, we decided to conduct the additional direct replication experiment in which participants are required to verbally report the visual targets, as requested by the reviewer. In case of an approval of our submission based on the Stage-1 criteria, we intend to conduct the additional direct replication within the next 4-6 months. The final replication report will then contain (a) the direct replication (Experiment 1) and (b) the close replication using a visual recognition measure instead of verbal recall (Experiment 2), allowing direct comparison of the emotional effects obtained with the two different response procedures.

We thank you again for your consideration and look forward to your response.

Kind regards,
Florian Kattner and Alexandra Clausen

Appendix C

To:

Professor Chris Chambers

Cardiff University

21 August, 2020

Dear Dr Chambers,

We would like to submit a Stage-2 Replication manuscript entitled “Revisiting the prioritization of emotional information in iconic memory” (RSOS-191507) for publication in *Royal Society Open Society*.

As mentioned in our previous correspondence, we were unable to recruit the full sample size of $N = 68$ participants for Experiment 1 due to the Covid-19 pandemic, and we are thus presenting results based on $N = 41$ participants prior to the outbreak. We also included additional bootstrapping analyses to demonstrate the robustness of our findings.

Apart from this approved deviation, we strictly followed the pre-registered protocol with regard to the experimental procedures and statistical analyses. The data of both experiments as well as the analysis scripts are accessible on the OSF project page. In the current manuscript, we have included a link referring to the accepted pre-registered protocol (p. 8) as well as a view-only link to the data (p. 34):

<https://osf.io/4hgcz>

https://osf.io/kj69f/?view_only=ebe5fdbaf029465ab01bdac9ee2f445d

We hope that the outcome of this Stage-2 Replication and our manuscript will meet your and the reviewers’ expectations, and we look forward to hearing from you.

Sincerely,

Florian Kattner and Alexandra Clausen

Appendix D

To:
Professor Chris Chambers
Cardiff University
Royal Society Open Science

13 September, 2020

Dear Dr Chambers,

We thank you for accepting our Stage-2 Replication manuscript “Revisiting the prioritization of emotional information in iconic memory” (RSOS-191507.R1) for publication in *Royal Society Open Science* subject to minor revisions.

We highly appreciate your and the reviewer’s time and effort to review the manuscript and to provide additional suggestions which we have now addressed as follows.

- The raw data of both experiments have been uploaded to the OSF archive as .csv files.
- On p. 9 we now mention explicitly that we used the identical set of twelve drawings as in the original study (we thank the first author for sharing the materials in the acknowledgement).
- We also agree with the reviewer that the relatively large differences in recall and recognition performance within each valence category point to a possible limitation of the present two experiments as well as the original study. We have thus expanded our discussion of this observation on p. 30, explicitly noting that more research may be required to test whether the effects on iconic memory were driven by stimulus properties other than the emotional meaning.

Hoping that these revisions will meet your expectations, we look forward to hearing from you.

Kind regards,

Florian Kattner and Alexandra Clausen